# Prediction and Compensation of Color Deviation by Response Surface Methodology for PolyJet 3D Printing

**Xingjian Wei** [1,*] , **Abhinav Bhardwaj** [1] , **Li Zeng** [2] **and Zhijian Pei** [1]

1    Department of Industrial & Systems Engineering, Texas A & M University, College Station, TX 77843, USA; abhinavb@tamu.edu (A.B.); zjpei@tamu.edu (Z.P.)
2    School of Data Science, City University of Hong Kong, Kowloon, Hong Kong, China; li.zeng@cityu.edu.hk
*    Correspondence: we.iko.ken@tamu.edu

**Abstract:** PolyJet 3D printing can produce any color by mixing multiple materials. However, there are often large deviations between the measured color of printed samples and the target color (when the target color is used as the specified color in the printer software). Therefore, to achieve a target color on a printed sample, the specified color in the printer software should not be the same as the target color. This study applies response surface methodology (RSM) to determine the optimal color specification to compensate for color deviations of the measured color of printed samples from the target color in PolyJet 3D printing. The RSM has three steps. First, a set of experiments are designed for a target color according to central composite design. Second, the experimental data are used to develop a second-order multivariate multiple regression model to predict the deviation between the measured color and the target color. Third, the optimal color specification (often different from the target color) is determined by using the developed predictive model and the desirability function. When the optimal color specification is used as the specified color in the printer software, the deviation between the predicted color of the printed sample and the target color is minimized. The proposed method is applied to four target colors to demonstrate its effectiveness. The results show that the proposed method performs better than the conventional color specification method without compensation in achieving the four target colors by 33% on average.

**Keywords:** 3D printing; color deviation; design of experiments; multivariate multiple regression; PolyJet; response surface methodology

## 1. Introduction

Three-dimensional printing provides a quick way to fabricate objects from computer-aided design (CAD) files. In the past decades, it has demonstrated its advantages over traditional manufacturing methods for complex and customizable parts [1]. Color 3D printing has become possible mainly thanks to the introduction of multi-material printing. The color of 3D printed objects had been limited to the color of the base material itself in the past, and now any color can be theoretically produced by mixing multiple base materials. It is desirable to have the color of a 3D printed object as close to the target color as possible for several reasons. For example, when 3D printing is used to produce medical models [1], it is desirable that 3D printed anatomies have the color as close to real anatomies as possible. Such precise representations of colors could enhance the effectiveness of medical models used in surgical planning and medical education. In addition, when 3D printing is used for product prototyping, the ability to precisely represent the color of 3D printed objects is needed to determine the final color of designed products used for mass production. Furthermore, when 3D printing is used for making final products, the color of printed products has psychological effects on the users' perception, and an inaccurate color representation can significantly change the attractiveness and impression of the products.

PolyJet 3D printing can produce any color on a single object [2,3]. Although mechanical properties of parts printed from the PolyJet 3D printing process may not be sufficiently

high for load-bearing applications, the process has its advantages in certain applications such as medical models or product prototyping [1]. However, every printer has its own achievable range of colors and tones [4]. As a consequence, the color of the printed object may vary from printer to printer when the same target color is used as the specified color in the printer software. The authors' preliminary experimental data, summarized in Table 1, show that the measured color of a printed sample often exhibits considerable deviations from the target color (when the target color is used as the specified color) in the PolyJet printer software. The colors in Table 1 are presented using the RGB color system. In the RGB color system, a color can be represented by three integer numbers R, G, and B, each ranging from 0 to 255. For example, the black color in the RGB color system is (0, 0, 0), and the white color is (255, 255, 255). Here, the measured color refers to the RGB values measured by a color measurement device described in Section 2.1.5, and the specified color is defined as the RGB values entered by the user in the printer software. Ideally, measured RGB values of printed samples should match the specified RGB values in the software. However, because the measured RGB values are often different from the specified RGB values, as shown in Table 1, the printed samples will not have the target color (if the target color is used as the specified color).

**Table 1.** Comparison of specified RGB values and measured RGB values.

| Specified or Measured Color | Colors of Four Printed Samples | | | |
|---|---|---|---|---|
| | (100, 100, 100) | (175, 75, 75) | (75, 175, 75) | (75, 75, 175) |
| Specified color (R, G, B) | | | | |
| | (109, 108, 102) | (166, 84, 80) | (94, 170, 73) | (84, 82, 146) |
| Measured color (R, G, B) | | | | |

Literature reviews on color 3D printing pointed out that surface color of 3D printed parts should be studied as much as other aspects of 3D printing such as material formulation, microstructure optimization, and mechanical properties, to meet modern aesthetic and practical standards [5,6]. Color accuracy issues of six types of 3D printing processes are summarized by Yuan et al. [6]. The significance of colors in 3D printing is reflected in the large number of reported studies focusing on the colors of 3D printing [7–15]. Reported studies about the colorimetry of 3D printing focused on the effects of surface texture and printing orientation on the color appearance of 3D printed objects. Wang et al. reported that post-processing of printed samples in a powder-based color 3D printing process led to higher saturation and smaller chromatic aberration, and soaking the printed samples increased surface brightness [7]. Sun and Lai compared the glossy and matte finish objects captured by an RGB camera and established a regression model to estimate their color differences [8]. Xiao and Brainard studied the effects of surface texture (i.e., glossy finish vs. matte finish) on the color perception of observers. Their results showed that changing surface texture would not affect color appearance significantly [9]. Xiao et al. developed a color management system to minimize color variation among different printing orientations for powder-binder-based multi-jet printing (MJP) [10]. Sun and Sie also developed a method to improve color uniformity among different orientations in MJP [11]. Ludwig et al. reported that color uniformity was affected by printing orientation and post-processing [12]. Morovic et al. controlled composition of individual voxels to co-optimize both color accuracy and mechanical properties [13]. Wittbrodt and Pearce studied the effects of filament colors on mechanical properties in the fused deposition modeling

process and found that PLA filaments of five colors (Black, Gray, Blue, White, and Natural) resulted in distinctive ultimate tensile strength, yield strength, and maximum strain [14]. Eiriksson et al. studied relationships between color input and output spaces and predicted color output by building a Look-Up Table [15]. There are no reported investigations on deviations between the measured colors of 3D printed objects and target colors (when the target colors are used as the specified colors in the printer software). There are no commonly accepted methods to compensate for such deviations, either.

This paper will fill this gap in the literature by presenting, for the first time, a systematic approach (versus the trial-and-error approach) to find the optimal color specifications for the printer software to result in a color on the printed sample that has the smallest deviation from the target color. It reports a study on the color deviation problem in PolyJet printing using the response surface methodology (RSM). As a method for process optimization, RSM has been widely used [16]. Essentially, RSM can be broken down into three steps: (1) collecting experimental data of response(s) obtained by adjusting process parameters in close proximity to the maximum/minimum of the response(s), (2) developing a predictive model for the response(s) given the process parameters, and (3) optimizing the process parameters that predict the maximum/minimum of the response(s). This study follows these three steps. First, a set of designed experiments according to central composite design is conducted for a target color to collect measured RGB values of printed samples given specified RGB values in the printer software. The set of experiments have 15 different specified RGB values (the target RGB value plus 14 RGB values that are in close proximity to the target RGB value). Secondly, using the experimental data, a statistical predictive model is developed for predicting deviations between measured RGB values and the target RGB values when different specified RGB values are used in the printer software. Finally, the predictive model and the desirability function are used to find the optimal color specification (i.e., specified RGB values) such that the deviations between the predicted RGB values of a printed sample and the target RGB values are minimized. Four cases with four different target colors are used to demonstrate the effectiveness of the proposed method.

The rest of the paper is organized as follows. Section 2 presents the experimental setup and procedure, describes the methodology to establish the second-order multivariate multiple regression model, and reports the determination of the optimal color specification using the predictive model and the desirability function. Section 3 examines the effectiveness of the proposed method in four cases (four different target colors). The results obtained by the proposed method are compared with the performance of the conventional color specification method without compensation. Finally, Section 4 summarizes conclusions, and discusses the limitations of the proposed method and directions of future research.

## 2. Material and Methods

### 2.1. Experimental Setup and Procedure

2.1.1. PolyJet 3D Printer

The Stratasys J750 3D printer (Eden Prairie, MN, USA) is used to conduct the experiments. The PolyJet printing process is schematically illustrated in Figure 1a, and the J750 printer is shown in Figure 1b. J750 distinguishes itself from its predecessors by increasing material capacities from 3 to 6 types, enabling it to theoretically print 500,000 different colors [3]. In the PolyJet printing process, two types of photocurable resins in liquid form, i.e., base model material and support material, are selectively deposited from the print heads. The base model material is used to construct the object, while the support material is used to temporarily build foundations for the base model material to be printed on top of it. After a layer is cured by UV lights attached to both sides of the print heads, the build platform goes down by the height of one layer, and the materials are deposited again. These steps are repeated until the object is completed.

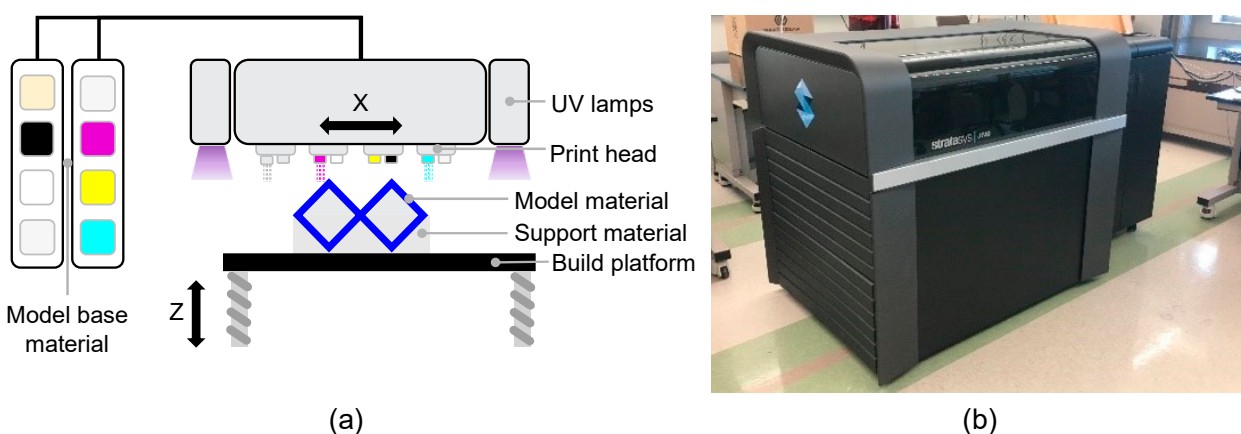

**Figure 1.** PolyJet 3D printing. (**a**) Illustration of the PolyJet printing process, and (**b**) picture of the Stratasys J750 Poly-Jet printer.

### 2.1.2. Process Parameters Related to Color and Their Settings

Important process parameters related to color in PolyJet printing include finish type, material combinations, and color profiles. There are two options for finish type: glossy finish and matte finish. When glossy finish is selected, support materials only cover the bottom surface of the object and the surface of overhanging structure. When matte finish is selected, support materials cover the entire surface of the object. Glossy finish type is used in this study to avoid potential problems caused by differences in removal of support material on the surfaces [17].

For material combinations, five base materials are mixed to generate a desired color: VeroPureWhite (RGD837), VeroBlackPlus (RGD875), VeroCyan (RGD843), VeroYellow (RGD836), and VeroMagenta (RGD851). SUP706B is used as the support material.

A profile in terms of printing color is a criterion to convert the RGB values used in digital monitors to CMYK (abbreviation for cyan, magenta, yellow, and black) values used in physical printers. Because the CMYK system does not cover the full spectrum of colors of the RGB system, a profile dictates how the software approximates RGB values to CMYK values [18]. "Natural texture" and "Natural shells" are chosen as the texture profile and color profile, respectively. There are two types of profiles because of two different options to print colors: the texture mapping-based option and the shell-based option. The texture mapping-based option that imposes color images on the surfaces of an object (its interior is white) follows the texture profile, while the shell-based option which allows users to assign colors to an entire object follows the color profile. The shell-based option is used to set specified RGB values in this study.

### 2.1.3. Experimental Design Using Central Composite Design

Central composite design (CCD) is often used to design experiments for a second-order model because it can provide enough samples without using a full factorial design [16]. In this study, each target color requires a CCD to collect measured RGB values and specified RGB values in proximity to the target RGB values. Figure 2 illustrates the points of RGB values in this experimental design. The RGB system is represented by the three axes. The coordinates of each design point are its coded RGB values. The center point represents the target color. Each experiment contains only one sample for each design point because previous experiments showed that the differences in measured RGB values for replicated samples are negligible [19]. Thus, a total of 15 samples are printed for a target color.

### 2.1.4. Preparation of Printed Samples

Printed samples are 20 mm × 20 mm square plates with a thickness of 2 mm. The dimensions are selected because the minimum scanning area required for the color measurement device is 15 mm$^2$. The sample geometry is generated in Autodesk Fusion 360 (San

Rafael, CA, USA). Layer thickness of 27 μm is selected to save printing time. The samples are printed 15 months after the printer installation, and maintenances such as cleaning have been carried out according to the manufacturer's recommendations. Right before each printing, cleaning wizard, a standard procedure of the printer to reduce color contaminations on print heads, is performed. After the completion of printing, gel-like support material (SUP706B) is manually removed by using a scraper. Then, pressurized water is blasted to the samples for further removal of support materials.

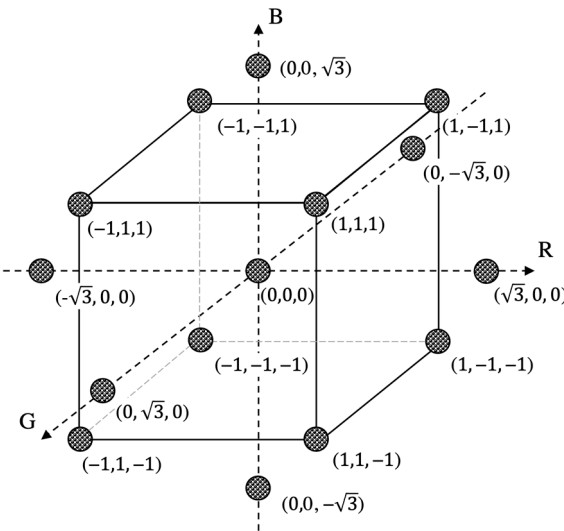

**Figure 2.** Illustration of a central composite design for coded RGB values.

2.1.5. Measurement of Color

The color of every printed sample is measured by a colorimeter, Nix Pro color sensor (Ontario, Canada). Illuminant and observer are two parameters that need to be set for the colorimeter [20]. The illuminant used is D50 which emulates horizon daylight with a color temperature of 5000 Kelvin [21]. The observer (the receiver of the illuminant) is set to 2° field of view, corresponding to the angle of cones in the fovea of human eyes. Three measurements are taken on each sample and the average of the three measurements yields the measured RGB values for the sample.

*2.2. Second-Order Multivariate Multiple Regression*

Using the experimental data obtained according to CCD, a second-order multivariate multiple regression (MMR) model can be built to predict the deviations between measured RGB values and target RGB values when different specified RGB values are used in the printer software. The second-order MMR model is suitable because there are multiple responses and predictor variables. Definitions of variables and coefficients involved in the MMR are summarized in Table 2. Assume there are $n$ observations in the training data. Each observation contains 6 elements $\{x_{i1}, x_{i2}, x_{i3}, y_{i1}, y_{i2}, y_{i3} : i = 1, 2 \ldots, n\}$, where the former three are the specified RGB values and the latter three are the measured RGB values. In the MMR model, deviations of measured RGB values from the target RGB values are treated as responses and the specified RGB values are treated as predictors. The predictive model for the $k$-th, $k = 1, 2, 3$, response is as follows

$$y_{ik} - t_k = \beta_{0k} + \sum_{j=1}^{3} \beta_{jk} x_{ij} + \sum_{l,q} \beta_{lqk} x_{il} x_{iq} + \sum_{j=1}^{3} \beta_{jjk} x_{ij}^2 + \varepsilon_{ik} \tag{1}$$

where $\beta_{0k}$ is the intercept, $\beta_{jk}$ is the coefficient of the first-order term, $\beta_{lqk}$ is the coefficient of the interaction term, $\beta_{jjk}$ is the coefficient of the second-order term, and $\varepsilon_{ik}$ is the random

noise in Equation (1). The coefficients of the model can be estimated by the least-squares method [16] using experimental data (i.e., training data).

**Table 2.** Definitions of variables and coefficients.

| Variable/Coefficient | Definition |
|---|---|
| $i$ | Index of observation in training data, $i \in \{1, \ldots, 15\}$ |
| $j$ | Index of predictor, $j \in \{1, 2, 3\}$ |
| $l$ | Index of predictor, $l \in \{1, 1, 2\}$ |
| $q$ | Index of predictor, $q \in \{2, 3, 3\}$ |
| $k$ | Index for response, $k \in \{1, 2, 3\}$ |
| $x_{ij}$ | $j$-th predictor of $i$-th observation |
| $x_{il}$ | $l$-th predictor of $i$-th observation |
| $x_{iq}$ | $q$-th predictor of $i$-th observation |
| $x_j$ | $j$-th given predictor |
| $x_j^*$ | $j$-th optimal specified predictor |
| $y_{ik}$ | $k$-th response of $i$-th observation |
| $y_k$ | $k$-th measured response for the given predictor |
| $t_k$ | $k$-th target response |
| $\beta_{0k}$ | Regression intercept for $k$-th response |
| $\beta_{jk}$ | First-order regression coefficient of $j$-th predictor for $k$-th response |
| $\beta_{lqk}$ | Two-way interaction coefficient between $l$-th predictor and $q$-th predictor for $k$-th response |
| $\beta_{jjk}$ | Second-order regression coefficient of $j$-th predictor for $k$-th response |
| $A$ | Maximum acceptable difference between predicted response and target response |
| $d_k$ | Individual desirability of $k$-th response |
| $D$ | Overall desirability |

Once the MMR model is established, it can be used to predict the deviations between measured RGB values of printed sample and target RGB values (when different specified RGB values are used in printer software). Let the $k$-th specified RGB values be $\{x_1, x_2, x_3\}$, and the corresponding measured RGB values be $\{y_1, y_2, y_3\}$. The prediction of the deviation of the $k$-th response is

$$\hat{y}_k - t_k = \hat{\beta}_{0k} + \sum_{j=1}^{3} \hat{\beta}_{jk} x_j + \sum_{l,q} \hat{\beta}_{lqk} x_l x_q + \sum_{j=1}^{3} \hat{\beta}_{jjk} x_j^2 \tag{2}$$

where the "ˆ" sign of the coefficients indicates their estimates based on the training data.

*2.3. Determination of the Optimal Color Specification*

The optimal color specification (i.e., the specified RGB values to be used in the printer software aiming to achieve the measured RGB values of printed sample as close to the target RGB values as possible) can be determined by minimizing the difference between the predicted RGB values of printed sample and the target RGB values. Since it is necessary to optimize the specified R, G, and B values simultaneously, this is a multi-response optimization problem. Desirability function is a popular methodology for multi-response optimization [22]. The optimal color specification is defined to be the specified RGB values $\{x_1^*, x_2^*, x_3^*\}$ whose corresponding predicted RGB values $\{\hat{y}_1^*, \hat{y}_2^*, \hat{y}_3^*\}$ from Equation (2) are closest to the given target RGB values $\{t_1, t_2, t_3\}$. In other words, the differences between $\{\hat{y}_1^*, \hat{y}_2^*, \hat{y}_3^*\}$ and $\{t_1, t_2, t_3\}$ are minimized. Since there are three responses, three individual desirability functions are firstly defined to measure the difference for each response. Specifically, the individual desirability function for the $k$-th, $k = 1, 2, 3$, response is defined as

$$d_k = \begin{cases} 1 - \frac{|\hat{y}_k^* - t_k|}{A} & if \ A \geq |\hat{y}_k^* - t_k| \\ 0 & otherwise \end{cases} \tag{3}$$

where $A$ is the acceptance range set by the user. Intuitively, the desirability $d_k$ indicates the difference between the predicted RGB values and their corresponding target RGB values in Equation (3). If this difference is larger than the acceptance range, the desirability is 0, meaning that it is not acceptable. When the difference is within the acceptance range, the desirability is a linear function of the difference, and increases as the difference decreases. The highest desirability, i.e., $d_k = 1$, is achieved when the difference is 0, that is, the predicted RGB values exactly match the target RGB values. The overall desirability function as the objective function in the optimization step is a multiplication of the three individual desirability functions in Equation (4).

$$D = \prod_{k=1}^{3} d_k \tag{4}$$

The optimal color specification is then obtained by maximizing the overall desirability in Equation (5).

$$\{x_1^*, x_2^*, x_3^*\} = \underset{\{x_1, x_2, x_3\}}{\arg\max} D \tag{5}$$

## 3. Demonstration of the Effectiveness of the Proposed Method by Four Cases

Demonstration of the effectiveness of the proposed method consists of four steps: (1) conducting a set of designed experiments according to the central composite design (CCD) with the target color as its center point, (2) developing a second-order multivariate multiple regression model using the experimental data (training data), (3) finding the optimal color specification using the developed model and the desirability function, and (4) printing the sample using the optimal color specification as the specified color in the printer software. The proposed method needs to be applied to every target color because the model's predictions are expected to be effective in the close proximity of the target color. Four target colors are selected to demonstrate the effectiveness of the proposed method. The four target colors are: (1) a color with equal RGB values, (2) a color with a larger red value (meaning that the R-value is much higher than the G and B values), (3) a color with a larger green value, and (4) a color with a larger blue value.

### 3.1. Conducting a Set of Designed Experiments and Collecting Experimental Data

For each target color, 15 samples are printed according to CCD, and are shown in Figure 3. Tables A1–A4 show the CCD matrices and measurement results for Case 1, Case 2, Case 3, and Case 4, respectively. It can be observed that there are considerable deviations between the measured RGB values and the target RGB values.

### 3.2. Developing Multivariate Multiple Regression (MMR) Model

For each of the four target colors, the second-order MMR model as described in Equations (1) and (2) is fit using the experimental data. Table A5 shows the coefficient estimates of the model for each of the four cases. These models can be used to predict the deviation of measured RGB values from the target RGB values. Two observations can be made. First, the non-zero coefficient estimates of the three first-order terms in all the responses of the four cases indicate that each of the measured RGB values depends on all three specified RGB values, instead of merely depending on one of the specified RGB values. Second, the coefficient estimates of the interaction and second-order terms are not negligible, which indicates the presence of interaction and second-order effects.

To provide a visual presentation of the MMR models, Figure 4 displays the deviation response surface plots of Case 3 predicted by the MMR model. Only two of the specified RGB values can be shown in each plot. First, it can be observed that the effects of the specified G value on the deviations are substantial. The deviations of the measured R values positively increase as the specified G value increases, and the deviations of the measured B value negatively increase as the specified G value increases. In fact, none of

the response surfaces are completely flat, which indicates all the specified RGB values have effects on the deviation. Second, there are also noticeable interaction and second-order effects of the specified RGB values on the deviation. For example, the response surface of the deviation of measured R-value based on the specified G and B values shows a convex curvature, and the response surface of the deviation of measured G value based on the specified R and G values shows a concave curvature, meaning that second-order effects exist. Moreover, the response surface of the deviation of measured R-value based on the specified R and G values shows a twisted plane, indicating that interaction effects are present.

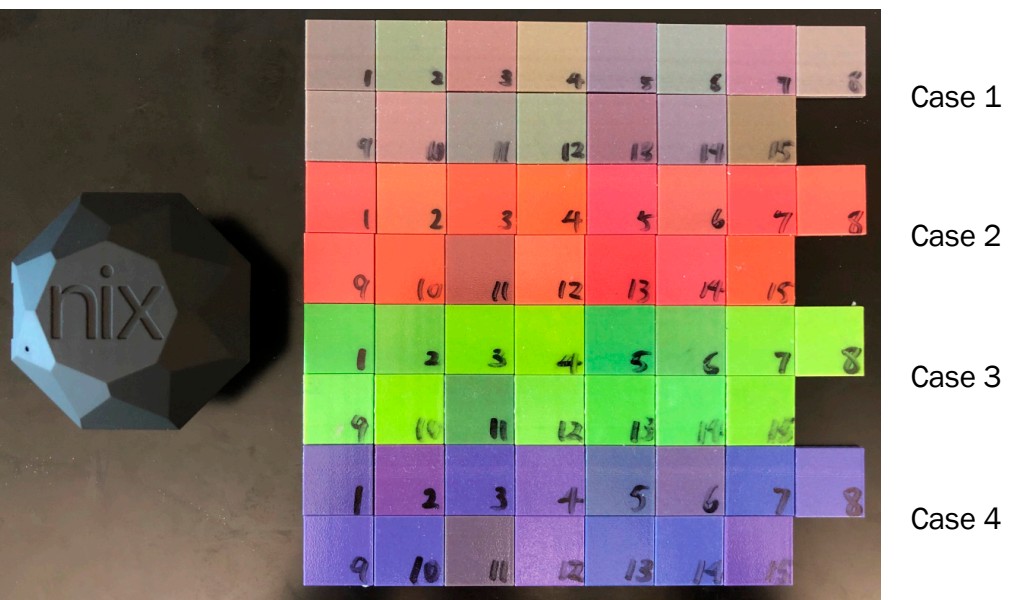

**Figure 3.** The Nix Pro colorimeter and printed samples for four cases.

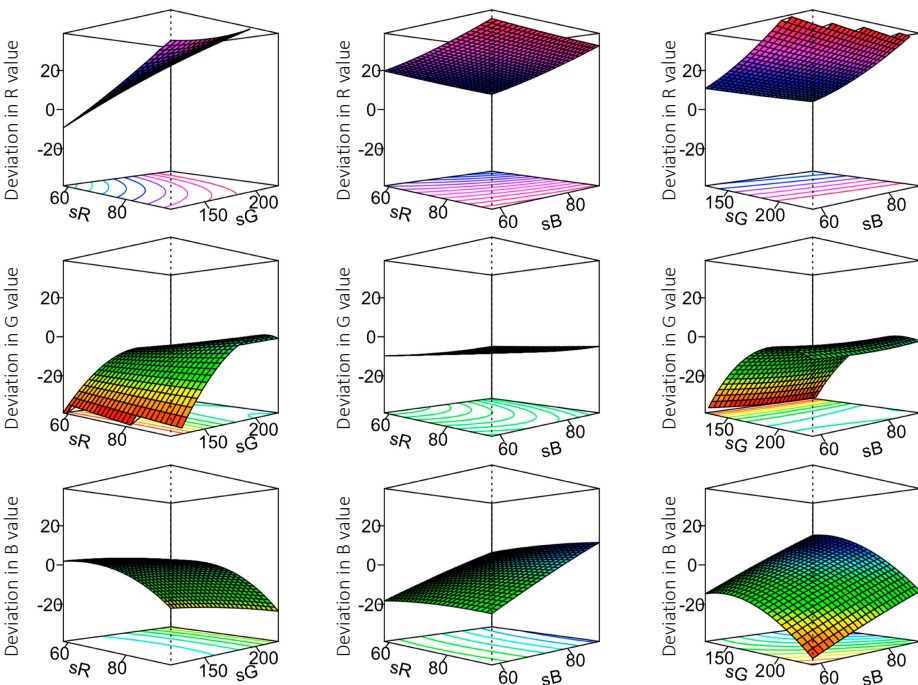

**Figure 4.** Plots of response surfaces for Case 3 based on the coefficient estimates (raw data are provided in Table A5).

### 3.3. Finding the Optimal Color Specification Using the Developed Model and the Desirability Function

Based on the developed MMR models, the optimal color specification for each target color is determined by the desirability function as represented by Equations (3)–(5). Table 3 shows the optimal color specifications (that should be used as specified RGB values to print samples) and the associated desirability values. The overall desirability values for Case 1 and Case 3 are close to their maximum, 1, meaning that the predicted RGB values are close to the target values. The overall desirability values for Case 2 and Case 4 are relatively low, suggesting that the predicted RGB values are not close to the target values.

**Table 3.** Optimal color specifications determined by individual and overall desirability values.

| Case | RGB | Desirability Individual | Desirability Overall | Target | Predicted | Optimal Color Specification |
|---|---|---|---|---|---|---|
| 1 | R | 0.99 | | 100 | 100 | 83 |
| | G | 0.92 | 0.95 | 100 | 100 | 92 |
| | B | 0.93 | | 100 | 100 | 97 |
| 2 | R | 0.33 | | 175 | 165 | 197 |
| | G | 0.66 | 0.60 | 75 | 78 | 68 |
| | B | 0.98 | | 75 | 75 | 76 |
| 3 | R | 0.96 | | 75 | 75 | 16 |
| | G | 0.96 | 0.96 | 175 | 175 | 178 |
| | B | 0.96 | | 75 | 75 | 95 |
| 4 | R | 0.90 | | 75 | 76 | 59 |
| | G | 0.31 | 0.34 | 75 | 82 | 72 |
| | B | 0.14 | | 175 | 145 | 198 |

### 3.4. Printing Samples Using the Optimal Color Specifications

To show the advantage of the proposed method over the conventional color specification method without compensation (i.e., using the target color as the specified color), two samples are printed for each target color. One sample is printed using the optimal color specification, determined by the proposed method, shown in Table 3 as the specified RGB values in the printer software, and another sample is printed using the target color as the specified RGB values in the printer software. Both samples are measured by the colorimeter.

Table 4 compares the measured RGB values obtained by the proposed method and the measured RGB values obtained by the conventional specification method without compensation for the four cases. Such comparison is also shown in Figure 5. The proposed method produces smaller deviations (in terms of the sum of deviations) than the conventional specification method without compensation in all four cases. The best results are in Case 1 where the sum of deviations from the proposed method is less than 40% of that from the conventional specification method without specification. This comparison demonstrates that the proposed method can achieve the target color better than the conventional specification method without compensation.

**Table 4.** Comparison of deviations of measured RGB values on printed samples from the target RGB values using the proposed method versus the conventional specification method without compensation.

| Case | RGB | Target | Proposed Method Specified | Proposed Method Measured | Proposed Method Deviation | Proposed Method Sum of Dev. | Conventional Specification Method Specified | Conventional Specification Method Measured | Conventional Specification Method Deviation | Conventional Specification Method Sum of Dev. |
|---|---|---|---|---|---|---|---|---|---|---|
| 1 | R | 100 | 83 | 95 | 5 | | 100 | 109 | 9 | |
| | G | 100 | 92 | 101 | 1 | 7 | 100 | 108 | 8 | 19 |
| | B | 100 | 97 | 101 | 1 | | 100 | 102 | 2 | |

**Table 4.** *Cont.*

| Case | RGB | Target | Proposed Method | | | | Conventional Specification Method | | | |
|------|-----|--------|-----------|----------|-----------|-------------|-----------|----------|-----------|-------------|
| | | | Specified | Measured | Deviation | Sum of Dev. | Specified | Measured | Deviation | Sum of Dev. |
| 2 | R | 175 | 197 | 166 | 9 | | 175 | 166 | 9 | |
| | G | 75 | 68 | 79 | 4 | 14 | 75 | 84 | 9 | 23 |
| | B | 75 | 76 | 74 | 1 | | 75 | 80 | 5 | |
| 3 | R | 75 | 16 | 77 | 2 | | 75 | 94 | 19 | |
| | G | 175 | 178 | 164 | 11 | 22 | 175 | 170 | 5 | 26 |
| | B | 75 | 95 | 84 | 9 | | 75 | 73 | 2 | |
| 4 | R | 75 | 59 | 76 | 1 | | 75 | 84 | 9 | |
| | G | 75 | 72 | 83 | 8 | 42 | 75 | 82 | 7 | 45 |
| | B | 175 | 198 | 142 | 33 | | 175 | 146 | 29 | |

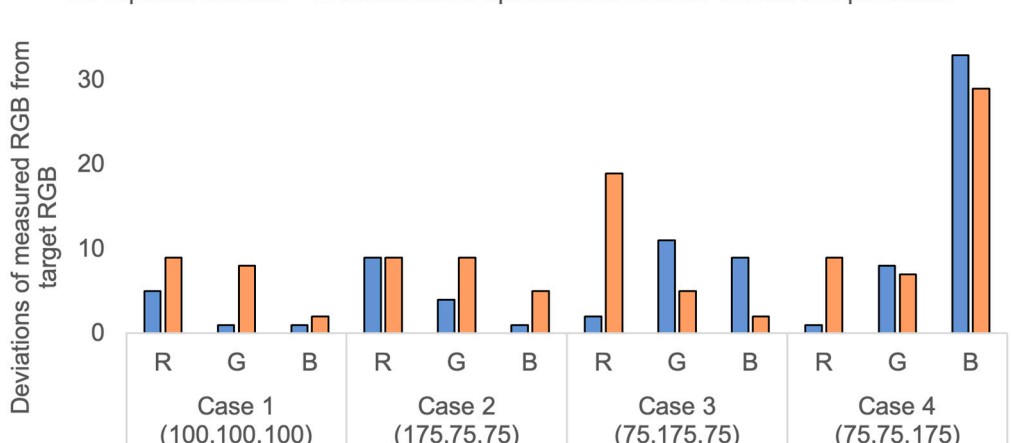

**Figure 5.** Deviations of measured RGB values from target RGB values when using the proposed method versus the conventional specification method without compensation.

## 4. Conclusions

A method (based on the response surface methodology) was proposed to compensate for the color deviation of a printed sample from the target color for the printed sample in PolyJet 3D printing. The key of the study was to use the optical color specification (not the target color) determined by the proposed method as the specified color in the printer software. The proposed method consisted of three steps: (1) collecting experimental data of printed samples according to the central composite design with its center being the target color, (2) developing a second-order multivariate multiple regression model using the experimental data to predict the deviation of measured color from the target color, and (3) finding the optimal color specification (that minimizes the color deviation when used as the specified color in the printer software) using the developed model and the desirability function.

This method was applied to four cases (each has a different target color) to demonstrate its effectiveness. Experimental results showed that the proposed method performed better than the conventional specification method without compensation (directly using the target color as the specified color in printer software) in all four cases (four target colors). The average improvement over the conventional specification method without compensation was 33%. It is worth mentioning that the proposed compensation method was not intended to cover the entire color gamut. In order for the compensation method to work, experiments had to be conducted in proximity to the target color. This paper presented a systematic approach (versus the trial-and-error approach) to find the optimal color specifications for the printer software to result in a color on the printed sample that has the smallest deviation from the target color. Even though this study focused on the PolyJet printing

process, the proposed method can also be applied to any 3D printing process to improve its color accuracy.

Future investigations are needed to understand why there are performance differences of the proposed method among the four cases. One approach towards consistent performance is to use advanced statistical predictive methods such as neural networks and Gaussian process to improve model predictive capacity. Furthermore, understanding whether and how the printing parameters, such as finish type and layer thickness, affect the color deviation of printed samples could further improve the explanatory power of the predictive methods.

**Author Contributions:** Conceptualization, X.W.; methodology, X.W. and L.Z.; validation, X.W. and A.B.; investigation, X.W.; writing—original draft preparation, X.W.; writing—review and editing, A.B., L.Z. and Z.P. All authors have read and agreed to the published version of the manuscript.

**Funding:** This research received no external funding.

**Conflicts of Interest:** The authors declare that they have no known competing financial interests or personal relationships that could have appeared to influence the work reported in this paper.

## Appendix A

**Table A1.** CCD matrix and results for Case 1.

| Sample | Specified RGB Values | | | Coded RGB Values | | | Measured RGB Values | | |
|---|---|---|---|---|---|---|---|---|---|
| | sR | sG | sB | cR | cG | cB | mR | mG | mB |
| 1 | 90 | 90 | 90 | −1 | −1 | −1 | 99 | 97 | 92 |
| 2 | 90 | 110 | 90 | −1 | 1 | −1 | 100 | 115 | 92 |
| 3 | 110 | 90 | 90 | 1 | −1 | −1 | 117 | 97 | 93 |
| 4 | 110 | 110 | 90 | 1 | 1 | −1 | 118 | 116 | 93 |
| 5 | 90 | 90 | 110 | −1 | −1 | 1 | 102 | 99 | 113 |
| 6 | 90 | 110 | 110 | −1 | 1 | 1 | 104 | 119 | 112 |
| 7 | 110 | 90 | 110 | 1 | −1 | 1 | 119 | 98 | 113 |
| 8 | 110 | 110 | 110 | 1 | 1 | 1 | 119 | 118 | 112 |
| 9 | 100 | 100 | 100 | 0 | 0 | 0 | 110 | 107 | 101 |
| 10 | 117 | 100 | 100 | 1.73 | 0 | 0 | 125 | 106 | 103 |
| 11 | 83 | 100 | 100 | −1.73 | 0 | 0 | 105 | 108 | 102 |
| 12 | 100 | 117 | 100 | 0 | 1.73 | 0 | 110 | 123 | 103 |
| 13 | 100 | 83 | 100 | 0 | −1.73 | 0 | 110 | 91 | 103 |
| 14 | 100 | 100 | 117 | 0 | 0 | 1.73 | 111 | 107 | 119 |
| 15 | 100 | 100 | 83 | 0 | 0 | −1.73 | 109 | 105 | 83 |

**Table A2.** CCD matrix and results for Case 2.

| Sample | Specified RGB Values | | | Coded RGB Values | | | Measured RGB Values | | |
|---|---|---|---|---|---|---|---|---|---|
| | sR | sG | sB | cR | cG | cB | mR | mG | mB |
| 1 | 145 | 65 | 65 | −1 | −1 | −1 | 153 | 78 | 75 |
| 2 | 145 | 85 | 65 | −1 | 1 | −1 | 150 | 94 | 71 |
| 3 | 205 | 65 | 65 | 1 | −1 | −1 | 163 | 79 | 63 |
| 4 | 205 | 85 | 65 | 1 | 1 | −1 | 168 | 89 | 60 |
| 5 | 145 | 65 | 85 | −1 | −1 | 1 | 155 | 79 | 95 |
| 6 | 145 | 85 | 85 | −1 | 1 | 1 | 152 | 96 | 91 |
| 7 | 205 | 65 | 85 | 1 | −1 | 1 | 166 | 79 | 81 |
| 8 | 205 | 85 | 85 | 1 | 1 | 1 | 171 | 89 | 79 |
| 9 | 175 | 75 | 75 | 0 | 0 | 0 | 164 | 84 | 79 |
| 10 | 227 | 75 | 75 | 1.73 | 0 | 0 | 164 | 80 | 62 |
| 11 | 123 | 75 | 75 | −1.73 | 0 | 0 | 127 | 82 | 78 |
| 12 | 175 | 92 | 75 | 0 | 1.73 | 0 | 168 | 99 | 78 |
| 13 | 175 | 58 | 75 | 0 | −1.73 | 0 | 159 | 74 | 81 |
| 14 | 175 | 75 | 92 | 0 | 0 | 1.73 | 165 | 84 | 96 |
| 15 | 175 | 75 | 58 | 0 | 0 | −1.73 | 162 | 85 | 65 |

**Table A3.** CCD matrix and results for Case 3.

| Sample | Specified RGB Values | | | Coded RGB Values | | | Measured RGB Values | | |
|--------|------|------|------|------|------|------|------|------|------|
| | sR | sG | sB | cR | cG | cB | mR | mG | mB |
| 1 | 65 | 155 | 65 | −1 | −1 | −1 | 88 | 160 | 67 |
| 2 | 85 | 155 | 65 | 1 | −1 | −1 | 101 | 162 | 67 |
| 3 | 65 | 195 | 65 | −1 | 1 | −1 | 108 | 167 | 56 |
| 4 | 85 | 195 | 65 | 1 | 1 | −1 | 117 | 172 | 57 |
| 5 | 65 | 155 | 85 | −1 | −1 | 1 | 82 | 162 | 85 |
| 6 | 85 | 155 | 85 | 1 | −1 | 1 | 98 | 164 | 86 |
| 7 | 65 | 195 | 85 | −1 | 1 | 1 | 102 | 169 | 70 |
| 8 | 85 | 195 | 85 | 1 | 1 | 1 | 111 | 173 | 73 |
| 9 | 75 | 175 | 75 | 0 | 0 | 0 | 99 | 168 | 71 |
| 10 | 75 | 210 | 75 | 0 | 1.73 | 0 | 115 | 173 | 59 |
| 11 | 75 | 140 | 75 | 0 | −1.73 | 0 | 87 | 148 | 77 |
| 12 | 92 | 175 | 75 | 1.73 | 0 | 0 | 111 | 172 | 73 |
| 13 | 58 | 175 | 75 | −1.73 | 0 | 0 | 90 | 166 | 71 |
| 14 | 75 | 175 | 92 | 0 | 0 | 1.73 | 95 | 170 | 87 |
| 15 | 75 | 175 | 58 | 0 | 0 | −1.73 | 104 | 167 | 59 |

**Table A4.** CCD matrix and results for Case 4.

| Sample | Specified RGB Values | | | Coded RGB Values | | | Measured RGB Values | | |
|--------|------|------|------|------|------|------|------|------|------|
| | sR | sG | sB | cR | cG | cB | mR | mG | mB |
| 1 | 65 | 65 | 125 | −1 | −1 | −1 | 81 | 78 | 128 |
| 2 | 85 | 65 | 125 | 1 | −1 | −1 | 99 | 79 | 128 |
| 3 | 65 | 65 | 225 | −1 | −1 | 1 | 85 | 82 | 141 |
| 4 | 85 | 65 | 225 | 1 | −1 | 1 | 94 | 83 | 141 |
| 5 | 65 | 85 | 125 | −1 | 1 | −1 | 82 | 96 | 127 |
| 6 | 85 | 85 | 125 | 1 | 1 | −1 | 98 | 95 | 127 |
| 7 | 65 | 85 | 225 | −1 | 1 | 1 | 73 | 85 | 145 |
| 8 | 85 | 85 | 225 | 1 | 1 | 1 | 84 | 86 | 144 |
| 9 | 75 | 75 | 175 | 0 | 0 | 0 | 84 | 82 | 142 |
| 10 | 75 | 75 | 255 | 0 | 0 | 1.6 | 84 | 83 | 143 |
| 11 | 75 | 75 | 95 | 0 | 0 | −1.6 | 87 | 84 | 98 |
| 12 | 91 | 75 | 175 | 1.6 | 0 | 0 | 97 | 83 | 141 |
| 13 | 59 | 75 | 175 | −1.6 | 0 | 0 | 73 | 83 | 143 |
| 14 | 75 | 91 | 175 | 0 | 1.6 | 0 | 84 | 93 | 152 |
| 15 | 75 | 59 | 175 | 0 | −1.6 | 0 | 88 | 78 | 137 |

**Table A5.** Coefficient estimates of the fitted MMR models.

| Case | $\hat{y}_k$ | $\hat{\beta}_0$ | $\hat{\beta}_1$ | $\hat{\beta}_2$ | $\hat{\beta}_3$ | $\hat{\beta}_{12}$ | $\hat{\beta}_{13}$ | $\hat{\beta}_{23}$ | $\hat{\beta}_{11}$ | $\hat{\beta}_{22}$ | $\hat{\beta}_{33}$ |
|------|------|------|------|------|------|------|------|------|------|------|------|
| 1 | | −100.360 | −1.158 | 1.091 | 1.409 | −0.003 | −0.005 | 0.000 | 0.013 | −0.004 | −0.004 |
| 2 | | −144.596 | 1.956 | −1.460 | −0.470 | 0.007 | 0.001 | 0.000 | −0.006 | 0.003 | 0.003 |
| 3 | mR | 91.109 | 1.098 | 0.626 | −0.753 | −0.010 | 0.006 | 0.000 | 0.005 | 0.001 | 0.001 |
| 4 | | −36.047 | 0.362 | −0.704 | 0.500 | 0.000 | −0.004 | −0.006 | 0.006 | 0.010 | 0.000 |
| 1 | | −71.940 | −0.078 | 0.158 | 0.486 | 0.001 | −0.004 | 0.004 | 0.001 | 0.001 | −0.002 |
| 2 | | −54.287 | 0.755 | −0.161 | −0.536 | −0.005 | −0.001 | 0.001 | −0.001 | 0.011 | 0.004 |
| 3 | mG | −152.095 | −0.708 | 1.892 | −0.800 | 0.005 | −0.003 | 0.003 | 0.001 | −0.006 | 0.004 |
| 4 | | 4.533 | −0.819 | −0.343 | 0.336 | −0.003 | 0.001 | −0.007 | 0.006 | 0.015 | 0.000 |
| 1 | | 33.926 | −0.744 | −1.132 | 1.542 | 0.000 | −0.003 | −0.003 | 0.005 | 0.007 | 0.000 |
| 2 | | −85.195 | 0.995 | −0.656 | 0.335 | 0.001 | −0.001 | 0.001 | −0.003 | 0.001 | 0.005 |
| 3 | mB | −153.829 | 0.176 | 0.923 | 1.433 | 0.001 | 0.004 | −0.003 | −0.004 | −0.003 | −0.002 |
| 4 | | −100.584 | −0.095 | −1.577 | 1.205 | −0.001 | 0.000 | 0.002 | 0.001 | 0.010 | −0.003 |

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
