# Peer review of "Prediction and Compensation of Color Deviation by Response Surface Methodology for PolyJet 3D Printing"

_jmmp, doi:10.3390/jmmp5040131_

Round 1
Reviewer 1 Report
This manuscript reports a research of applying response surface methodology to determine the optimal color specification to compensate color deviations of the measured color of printed objects from targeting color. The MMR model provides a prediction of the specified colors which can improve the color accuracy. This is a very interesting topic and I recommend it for publication after a minor revision.
- Why do the printed 15 samples from each one case are different from each other? In my opinion, it should be very similar since they printed from the same printer and same model. If the printer can't provide stable printing, it will be difficult to verify the predicted color value in the experiment.
- Some printed samples using the optimal color specifications from MMR should be provided and compared with the sample using the conventional method.
Author Response
This manuscript reports a research of applying response surface methodology to determine the optimal color specification to compensate color deviations of the measured color of printed objects from targeting color. The MMR model provides a prediction of the specified colors which can improve the color accuracy. This is a very interesting topic and I recommend it for publication after a minor revision.
The authors appreciate the positive comments and recommendation from the reviewer.
Why do the printed 15 samples from each one case are different from each other? In my opinion, it should be very similar since they printed from the same printer and same model. If the printer can't provide stable printing, it will be difficult to verify the predicted color value in the experiment.
The authors agree with the reviewer that “it should be very similar since they printed from the same printer and same model. If the printer can't provide stable printing, it will be difficult to verify the predicted color value in the experiment.” The authors’ previous study [Ref 17] shows that the printer can produce very similar color for a given set of color specification. Based on the experimental results, no replicated samples are printed for each experiment condition in this study.
For each case, 15 samples were printed using 15 different color specifications in close proximity to the target color, using a central composite design shown from Table A1 to Table A4. In other words, for each case, 15 samples were printed under 15 different printing conditions (color specifications). Only one sample was printed under each printing condition.
Some printed samples using the optimal color specifications from MMR should be provided and compared with the sample using the conventional method.
Samples were printed using the optimal color specifications predicted from the MMR model. Results are presented in Table 4 and Figure 5 and are compared with the samples printed using the conventional method.
Reviewer 2 Report
In this paper, the authors do a an experimental study to determine the deviation in color mixing during the PolyJet process and use the results to derive an optimization model to correct for expected deviations. The paper is well-written and clear, but it is a very limited study with a fairly small contribution to the literature. It is rigorously done, but the scope and applicability of the paper is much more like a letter or technical brief, not a full research paper. See my detailed comments below. I recommend asking the authors for a major revision to address the issues before any further review.
1. As said previously, this study is rigorous but very limited in scope and probably should be presented in a letter or technical brief, not a research article. The color issue in PolyJet and several other processes is well-known but as far as I am aware, this is not really a major concern at the present time since it has little or no effect on the mechanical properties of the final parts.
2. The optimization model is interesting, but would you not have to create a new model for each individual machine, brand of machine, brand of resin, even each specific color of resin? Since there is some serious variance in the process, it seems that any correction of the color would only be valid for a very limited, specific set of circumstances. The authors should consider adding more discussion of how this process could be automated and/or built into the machine software. This is mentioned very briefly at the end of the paper, but not explored nearly enough to be convincing for the reader.
3. There is some discussion about how the different parameters and properties would affect the color patterns. I think the discussion should be reversed, where parameters and properties should be prioritized in favor of appearance. This would better justify spending research resources and journal space exploring the color issue itself.
4. One thing the authors should discuss is that material jetting processes (of which PolyJet is a member process) really cannot be used for actual manufacturing, beyond perhaps some small ceramic devices and master parts for casting processes. Since it will rarely or never be used for applications where it is vital that the color be perfect, more justification is needed for why the issue is a major concern to be addressed. Studies on design, material properties, and expanding the application range for PolyJet already in the literature could be discussed and used as evidence to better present the research in this paper.
5. The literature review is weak and expanding it to include color issues with other processes and studies about the effect of color mixing on mechanical properties would improve the presentation.
6. Second to last line in Section 5: Should be "power" not "powder"
Good luck with your revision!
Author Response
In this paper, the authors do a an experimental study to determine the deviation in color mixing during the PolyJet process and use the results to derive an optimization model to correct for expected deviations. The paper is well-written and clear, but it is a very limited study with a fairly small contribution to the literature. It is rigorously done, but the scope and applicability of the paper is much more like a letter or technical brief, not a full research paper. See my detailed comments below. I recommend asking the authors for a major revision to address the issues before any further review.
The authors appreciate the positive comments from the reviewer, including that “The paper is well-written and clear” and “It is rigorously done”. The authors also agree with the reviewer that “it is a very limited study.” This paper reports a study that applies response surface methodology (RSM) to determine the optimal color specification to compensate color deviations of measured color of printed samples from the target color in PolyJet printing. Currently, there are no compensation methods for color deviations in the literature.
The authors also respect the reviewer’s comment that “the scope and applicability of the paper is much more like a letter or technical brief, not a full research paper.” If the journal editor decides to publish this paper as a letter or technical brief, the authors are willing to comply.
1. As said previously, this study is rigorous but very limited in scope and probably should be presented in a letter or technical brief, not a research article. The color issue in PolyJet and several other processes is well-known but as far as I am aware, this is not really a major concern at the present time since it has little or no effect on the mechanical properties of the final parts.
The authors respect the reviewer’s comment that “this study is rigorous but very limited in scope and probably should be presented in a letter or technical brief, not a research article.” If the journal editor decides to publish this paper as a letter or technical brief, the authors are willing to comply.
The authors agree with the reviewer’s comment that “the color inaccuracy has little or no effect on the mechanical properties of the final parts”. However, the color of printed parts can be important for some applications.
The following sentences have been added in the revised paper (page 1):
For example, when 3D printing is used to produce medical models [1], it is desirable that 3D printed anatomies have the color as close to real anatomies as possible. Such precise representations of colors could enhance effectiveness of medical models used in surgical planning and medical education. In addition, when 3D printing is used for product prototyping, the ability to precisely represent the color of 3D printed objects is needed to determine the final color of designed products used for mass production. Furthermore, when 3D printing is used for making final products, the color of printed products has psychological effects on users’ perception, and an inaccurate color representation can significantly change the attractiveness and impression of the products.
2. The optimization model is interesting, but would you not have to create a new model for each individual machine, brand of machine, brand of resin, even each specific color of resin? Since there is some serious variance in the process, it seems that any correction of the color would only be valid for a very limited, specific set of circumstances. The authors should consider adding more discussion of how this process could be automated and/or built into the machine software. This is mentioned very briefly at the end of the paper, but not explored nearly enough to be convincing for the reader.
This is an excellent point. The authors are in agreement with the reviewer. The proposed compensation method is not intended to cover the entire color gamut. Instead, it provides a systematic approach (versus the trial-and-error approach) to find the optimal color specifications for the printer software to result in a color on the printed sample that has the smallest deviation from the target color.
The following sentences have been added in the conclusion of the revised paper (page 11):
It is noted that the proposed compensation method is not intended to cover the entire color gamut. In order for the compensation method to work, experiments have to be conducted in the proximity to the target color. This paper presents a systematic approach (versus the trial-and-error approach) to find the optimal color specifications for the printer software to result in a color on the printed sample that has the smallest deviation from the target color.
3. There is some discussion about how the different parameters and properties would affect the color patterns. I think the discussion should be reversed, where parameters and properties should be prioritized in favor of appearance. This would better justify spending research resources and journal space exploring the color issue itself.
The authors agree with the reviewer that “parameters and properties should be prioritized in favor of appearance”. The authors have published several papers (listed below) to study the effects of printing parameters on mechanical properties and dimensional accuracy. However, the color of printed parts can be important for some applications.
-
- Sanders, J., Wei, X.J., and Pei, Z.J., 2019, “Experimental investigation of PolyJet 3D printing process: effects of orientation and layer thickness on thermal glass transition temperature,” Proceedings of the ASME 2019 International Mechanical Engineering Congress and Exposition (IMECE 2019), Salt Lake City, Utah; November 11-14, 2019, IMECE2019‑
- Wei, X.J., Bhardwaj, A., Shih, C.C., Zeng, L., Tai, B., and Pei, Z.J., 2019, “Experimental investigation of Stratasys J750 PolyJet printer: effects of orientation and layer thickness on mechanical properties,” Proceedings of the ASME 2019 International Manufacturing Science and Engineering Conference (MSEC 2019), Erie, Pennsylvania; June 10-14, 2019, MSEC2019‑
- Wei, X.J., Bhardwaj, A., Zeng, L., and Pei, Z.J., 2019, “Experimental investigation of Stratasys J750 PolyJet printer: effects of finish type and shore hardness on surface roughness,” Proceedings of the ASME 2019 International Manufacturing Science and Engineering Conference (MSEC 2019), Erie, Pennsylvania; June 10-14, 2019, MSEC2019‑
- Wei, X.J., Thakare, K., Zeng, L., and Pei, Z.J., 2019, “Experimental investigation of Stratasys J750 PolyJet printer: effects of finish type and shore hardness on dimensional accuracy,” Proceedings of the ASME 2019 International Manufacturing Science and Engineering Conference (MSEC 2019), Erie, Pennsylvania; June 10-14, 2019, MSEC2019‑
- Thakare, K., Wei, X.J., and Pei, Z.J., 2019, “Dimensional accuracy in PolyJet printing: a literature review,” Proceedings of the ASME 2019 International Manufacturing Science and Engineering Conference (MSEC 2019), Erie, Pennsylvania; June 10-14, 2019, MSEC2019‑
The following sentences have been added in the revised paper (page 1):
For example, when 3D printing is used to produce medical models [1], it is desirable that 3D printed anatomies have the color as close to real anatomies as possible. Such precise representations of colors could enhance effectiveness of medical models used in surgical planning and medical education. In addition, when 3D printing is used for product prototyping, the ability to precisely represent the color of 3D printed objects is needed to determine the final color of designed products used for mass production. Furthermore, when 3D printing is used for making final products, the color of printed products has psychological effects on users’ perception, and an inaccurate color representation can significantly change the attractiveness and impression of the products.
4. One thing the authors should discuss is that material jetting processes (of which PolyJet is a member process) really cannot be used for actual manufacturing, beyond perhaps some small ceramic devices and master parts for casting processes. Since it will rarely or never be used for applications where it is vital that the color be perfect, more justification is needed for why the issue is a major concern to be addressed. Studies on design, material properties, and expanding the application range for PolyJet already in the literature could be discussed and used as evidence to better present the research in this paper.
The authors agree that the material jetting processes would not be suitable for, e.g., load-bearing applications where high mechanical properties of the final parts are needed. The PolyJet process can be used to print final parts in certain applications thanks to its advantages such as low surface roughness and multi-material capability.
The following sentences have been added in the introduction of the revised paper (page 2):
Although mechanical properties of parts printed from the PolyJet 3D printing process are not sufficiently high for load-bearing applications, the process has its advantages in certain applications such as medical models or product prototyping [1].
5. The literature review is weak and expanding it to include color issues with other processes and studies about the effect of color mixing on mechanical properties would improve the presentation.
The following two review papers on color 3D printing are included in the revised manuscript as Refs 5&6, and are cited on page 2 in the following sentence.
Some literature reviews on color 3D printing pointed out that surface color of 3D printed parts should be studied as much as other aspects of 3D printing such as material formulation, microstructure optimization, and mechanical properties, to meet modern aesthetic and practical standards [5,6]. Color accuracy issues of six types of 3D printing processes are summarized by Yuan et al [6].
Ref. 14 is added as a reference for reporting effects of process parameters on both color and mechanical properties. It is cited on page 2 in the following sentence.
Wittbrodt and Pearce studied effects of filament colors on mechanical properties in fused deposition modeling process, and found that PLA filaments with five colors (Black, Gray, Blue, White, and Natural) resulted in distinctive ultimate tensile strength, yield strength, and maximum strain [14].
6. Second to last line in Section 5: Should be "power" not "powder"
It is corrected in the revised manuscript.
Reviewer 3 Report
From the point of view of the content in the Abstract part, it is not recommended to introduce the way of organizing the program experiment. It is the subject of Chapter 2.
There are equations that are not explicitly referred to in the text, for example, equation 1 for example.
Position 14 of the bibliographic references is not in agreement with the work in my opinion and is at the same time a self-citation of one of the authors. The 1 and 17 position from the bibliographic references are self-citations, in my opinion.
The analysis of the chapters is missing, chapter 4 which refers to discussions.
Author Response
From the point of view of the content in the Abstract part, it is not recommended to introduce the way of organizing the program experiment. It is the subject of Chapter 2.
The authors do not fully understand this comment. Assuming this comment is about the organization of the abstract, the authors believe that it is easier to understand the concept of the paper by firstly presenting the methodology, and then demonstrating its effectiveness.
There are equations that are not explicitly referred to in the text, for example, equation 1 for example.
The equation numbers are added at the end of each equation. In the revised version, all equations are referred in the text.
Position 14 of the bibliographic references is not in agreement with the work in my opinion and is at the same time a self-citation of one of the authors.
Ref. 14 is removed as suggested by the reviewer.
The 1 and 17 position from the bibliographic references are self-citations, in my opinion.
The authors would like to use Ref. 1 (a review paper) to show the importance of color 3D printing, and Ref. 17 (Ref. 19 now) to substantiate why no replicated samples are printed for each condition (color specification) in this study.
The analysis of the chapters is missing, chapter 4 which refers to discussions.
The authors did not fully understand this comment. Assuming this comment is about the depth of discussion in Chapter 4, the authors intentionally did not provide much discussion on experimental results in this paper. The main reason is that they did not have explanations to all their experimental results. If they tried to explain them, it would be primarily speculations. However, the authors are conducting further research on the topic, aiming to provide understanding of the observed results. They will publish their new results in separate papers.
Round 2
Reviewer 2 Report
Dear Authors,
Thank you for your revision. I think the paper is much better, especially with the added discussion about the limitations of the study and the technology. However, I still think the paper would be most appropriate as a Communication paper (after reviewing the allowed paper types in JMMP) and not a full research paper. In the future, the authors may consider doing a similar study but examining either the design or material property data vs the color issue (or both), which could make a very strong research article.
For the current paper, if the authors are willing to change the paper type, I support acceptance of the paper with no further revisions.